# Undergraduate occupational therapy students' perceptions of their preparedness for splinting in hand rehabilitation: An exploratory study at the University of KwaZulu-Natal

Sizophila Gwala[1], Nompumelelo Mtshali[1], Sithembile Nene[1], Casey Stanley[1], Luther Lebogang Monareng[1,2]*

1 Department of Occupational Therapy, College of Health Sciences, University of KwaZulu-Natal, Durban, South Africa, 2 University College London, London, United Kingdom

* leboganglolo@gmail.com, monarengl@ukzn.ac.za

## Abstract

### Background

Hand rehabilitation, particularly splinting, is a key area in occupational therapy. However, existing literature suggests that students often feel inadequately prepared, particularly concerning splinting in hand rehabilitation. This highlights the need for further research and improvements. This research explored the perceptions held by undergraduate occupational therapy students regarding splinting in hand rehabilitation at the University of KwaZulu-Natal, with the aim of informing curriculum improvements. The novelty of this study lies in its focus on the University of KwaZulu-Natal, providing institution-specific insights that are currently underexplored in the literature, using Kolb's Experiential Learning Theory, to contextualise this study.

### Methods

This qualitative study gathered data from 3rd and 4th year occupational therapy students at the University of KwaZulu-Natal using purposive sampling. The Consolidated Criteria for Reporting Qualitative Research (COREQ): a 32-item checklist for interviews and focus groups was followed. The data were collected through semi-structured interviews using a piloted question guide and thematic analysis was used to analyse the data. Ethical clearance was received for this research.

### Results

This research comprised 16 participants: n = 5 males and n = 11 females, all of whom were undergraduates. The number of physical clinical fieldwork blocks completed by the participants ranged from two to three. The number of splints made during these blocks ranged from no splints to four or more. Four distinctive themes emerged: student

**Data availability statement:** All raw data files are available from Open Science Framework (OSF). The full reference is as follows: UKZN undergraduate research Raw Data. OSF; 2024. Available from: https://osf.io/yvd7r. Contact the corresponding author for further details.

**Funding:** The author(s) received no specific funding for this work.

**Competing interests:** The authors have declared that no competing interests exist.

readiness and preparedness, challenges in splinting, comparison of readiness in 3rd and 4th year, and suggested solutions to bridge the gaps.

## Conclusion

Despite acquiring theoretical knowledge, students felt underprepared and uncertain due to limited clinical opportunities, inadequate supervision, and minimal hands-on practice during fieldwork. This lack of confidence and competence may deter students from pursuing specialisation in hand rehabilitation. To address these gaps, students recommended strategies such as increasing practical time dedicated to splinting, introducing training earlier in the program, expanding the splinting curriculum to cover a broader range of splint types, and strengthening university support structures, which align with existing literature, that emphasises the crucial role of hands-on experience in fostering confidence and competence. These findings have implications for curriculum development and suggest the need for policy reforms prioritising clinical competency in undergraduate occupational therapy education.

## Background

*Hand therapy and hand rehabilitation are used interchangeably in this study.

According to the American Occupational Therapy Association (2020) [1], occupational therapy (OT) is a holistic healthcare profession that helps individuals of all ages overcome physical, emotional, and cognitive challenges that affect their ability to engage in daily activities. It utilises a client-centred approach to promote health and well-being through meaningful occupations such as Activities of Daily Living (ADLs), work, and leisure [2]. Frameworks like the International Classification of Functioning (ICF) by the World Health Organization (WHO) guide OTs in assessing clients' functioning and disability, providing a standardised language to evaluate impairments and activity limitations [3].

Hand rehabilitation is an essential area within OT that focuses on treating various hand injuries and conditions, such as burns, tendon and nerve injuries, and arthritis [4]. A crucial aspect of hand rehabilitation is splinting, a treatment modality that supports, immobilises, and protects the hand, ultimately promoting optimal healing [4]. Effective splinting requires specific skills and training to prevent complications, emphasising the necessity of comprehensive education and training [5].

In South Africa, OT undergraduate students, including those at the University of KwaZulu-Natal (UKZN), are required to complete at least 1,000 hours of clinical or practical experience [6]. However, existing literature acknowledges students' lack of preparedness for splinting. Yet, few studies have investigated this issue in-depth within the South African context, particularly at institution-specific levels, such as UKZN. Despite national curriculum standards, local challenges such as resource constraints and variable clinical fieldwork exposure create unique training experiences that may not be captured in broader studies. These challenges may directly influence students' confidence and competence in splinting, raising questions about



the effectiveness of current training models in adequately preparing them for clinical practice. This study contributes towards addressing this gap by exploring the specific perceptions of UKZN OT students on UKZN's hand rehabilitation content, offering institution-specific insights to inform both curriculum development and practical training strategies. Kolb's Experiential Learning Theory [7] is used to contextualise this study. In doing so, the study contributes to the limited body of knowledge on how contextual factors influence students' preparedness in splinting for hand rehabilitation.

### Research question

What are the perceptions of OT students on undergraduate splinting as a part of hand rehabilitation at the University of KwaZulu-Natal?

**Aim:**

The study aimed to investigate the perceptions held by undergraduate OT students regarding splinting in hand rehabilitation at the University of KwaZulu-Natal.

### Objectives

i. To explore the OT students' perceptions of their readiness and preparedness for splinting in clinical fieldwork.

ii. To identify students' recommended strategies for addressing deficiencies in the UKZN OT hand rehabilitation section, specifically splinting.

## Literature review

This literature review section examines the critical role of undergraduate OT training at UKZN in addressing training issues, such as the amount of time allocated to training. It emphasises the need for practical experience to bridge the gap between theory and application in splinting as part of hand rehabilitation, aligning it with Kolb's Experiential Learning Theory. Ultimately, the intention is to enhance healthcare access and improve the population's well-being.

### Hand rehabilitation in KwaZulu-Natal

The scope of hand rehabilitation in KwaZulu-Natal (KZN), South Africa, is marked by significant challenges, including a shortage of OTs in the public sector, limited studies on rehabilitation outcomes, and budget constraints [8]. The prevalence of hand injuries in KZN is substantial, with a need for comprehensive data on hand rehabilitation in the South African context.

South Africa's healthcare landscape faces several challenges, including the distribution of OTs, accessibility to rehabilitation services, research gaps in service outcomes, and budgetary constraints [9]. In 2022, South Africa's population was just over 62 million [10], with a significant portion (71%) relying on the public health sector for their healthcare needs [11]. In 2018, there were 5180 OTs in South Africa [9]. In KZN, there were 75 OTs spread across 45 district hospitals and 14 regional hospitals, serving a population of over 11 million.

More than 90% of OTs with over five years of experience in hand rehabilitation tend to move into the private sector due to barriers such as limited access to resources, physical environment, client follow-up up and high caseloads [8]. This trend has significant implications for rural areas in South Africa, where access to hand rehabilitation services within the public health sector is limited. In 2018, a shortage of OTs was evident in KZN, where the division needed 19 OTs but had at most six positions filled across various public healthcare sectors. This shortage harms the delivery of OT services within the public health sector. According to Ghela et al. (2022) [8], the uneven distribution of OTs is attributed to the limited number of studies conducted on the outcomes of rehabilitation services provided at various levels of the public health service system. This makes it challenging to assess the effectiveness of rehabilitation efforts and pinpoint areas that require

improvement. Ghela et al. (2022) [8] added that government budget constraints pose a significant challenge, affecting therapists' ability to utilise specific treatments, such as hand splinting, due to the high cost of required materials.

Human hands are highly complex structures as they are made up of numerous bones, joints, tendons, ligaments, muscles, nerves and blood vessels, as well as sensory receptors, that all work in a coordinated way to produce precise, controlled movements that allow individuals to function and complete a variety of daily tasks [4,12]. Hand rehabilitation is a multifaceted process that requires a multidisciplinary approach to provide holistic treatment and optimal recovery and function. Although many professionals form part of the multidisciplinary team (MDT) for hand therapy, this has been an area of speciality for OTs since the 1940s [12]. Hand surgeons, nurses, and physiotherapists form part of the MDT to help maximise the outcomes. OT hand rehabilitation starts with a referral, evaluation, intervention, and follow-up with the client [13]. OTs address physiological issues using modalities such as splinting, manual techniques, and occupation-based activities, which are tailored to the client's goals [4,14].

Splinting as a part of hand rehabilitation within OT intervention addresses the biomechanical, functional and psychological complications related to hand injuries [4,14]. Hand rehabilitation aims to restore functional use by applying knowledge from physiology, anatomy, and psychology [4,14]. While physiology and anatomy guide the physical restoration of structures, psychology plays a critical role in addressing the emotional, cognitive, and behavioural factors that influence recovery. Following a hand injury, individuals may experience a wide range of emotional distress, including anxiety, depression, frustration, fear, and guilt [15]. The loss of hand function can also lead to feelings of insecurity, dependency, and perceived incompetence, further contributing to psychological strain [16]. Studies on work-related hand injuries have shown high incidences of stress-related conditions such as anxiety disorders, major depression, and post-traumatic stress disorder [17,18]. These psychological factors can negatively impact motivation, pain perception, coping ability, and ultimately, engagement and outcomes in therapy.

Traumatic hand injuries remain a significant global health concern [19], while just above 30% of traumatic injuries are hand-related in South Africa [20]. [13] However, comprehensive data regarding the prevalence of hand injuries in South Africa remains limited. Naidoo et al. (2021) [13] noted that the most frequently treated injuries and conditions included bone and joint conditions, arthritic conditions, tendon injuries, peripheral nerve injuries, complex injuries, thermal injuries, and infections, many of which resulted from accidents or other traumatic events. Among these, the most common traumatic hand injuries managed by OTs were flexor tendon injuries (88%), fractures (83%), extensor tendon injuries (73%), and combined hand injuries (73%). Vascular injuries were the least severe condition managed by OTs, accounting for only 17% of cases that year [13].

To address the issues highlighted above, it is crucial to invest in OT training and encourage the deployment of OTs within the public health sector. Strengthening the OT workforce will enhance healthcare access, improve rehabilitation outcomes, and ultimately, promote the well-being of South Africa's diverse population.

## Undergraduate occupational therapy training in hand rehabilitation

In South Africa, eight universities offer undergraduate programs in OT. In the province of KZN, UKZN at the Westville Campus is the sole institution offering this degree. The Health Professions Council of South Africa (HPCSA) mandates that the OT curriculum at each university should encompass various components, such as, but not limited to, 10% on understanding human structures and functions (anatomy and physiology), 10% on understanding diseases, disorders, and trauma (physiology and clinical sciences), 35% on occupation and OT, 10% on primary health care, promotion, community or social development, and social determinants of health [6]. The comprehensive OT curricula in South African universities provide a solid foundation for students to acquire the knowledge and skills necessary for effective patient care, including hand rehabilitation, and prepare them for professional practice.

This comprehensive knowledge base is crucial for students as it forms the foundation for their practical application during clinical fieldwork. UKZN's curriculum is designed to build upon this knowledge each year. For instance, students

learn about anatomy and physiology in their first and second years. These two subjects expose students to the fundamental knowledge about the structure and function of the hand. Their OT-specific sections begin in the second year, where students are introduced to various hand conditions and learn how to assess and conduct essential treatments. Different tools and materials are used for hand-related teaching, encompassing traditional lectures with slides, demonstration videos, recommended textbooks, and recent, contextually relevant published articles. The in-class hand lectures duration ranges from half a day (5 hours) to three full days (8 hours per day). In their 3rd and 4th year, the focus delves into splinting as an intervention modality, specifically learning about splinting and how to make splints. A splint is "a rigid or flexible material used to protect, immobilise, or restrict motion in a part" [21], p98. During the practical training sessions, students develop their clinical skills by fabricating hard and soft splints for adult and paediatric clients, practising on classmates' typical, uninjured hands.

This is an essential part of the learning process, as by practising on a typical, uninjured hand, students can build their confidence, become familiar with the tools and materials used and develop techniques without having the added complexity of an injury. Some splints that are explored and/or made are forearm-based splints, such as wrist extension splints, resting splints, and dynamic extension splints. Other splints introduced at the undergraduate level include hand-based splints, such as finger splints, thumb splints, and buddy strapping [22].

To enhance their skills, students are expected to attend clinical fieldwork and utilise opportunities, such as working in an MDT for applied learning at various fieldwork sites, including Substance Abuse Rehabilitation Centres, Sheltered Workshops, Old Age Homes, Chronic Care Facilities, Public Hospitals, Rehabilitation Centres, Community-based Centres, Special Needs Schools and other relevant settings. The four-year Occupational Therapy programme at UKZN structures its clinical fieldwork as 'blocks' – i.e., periods of supervised practice that begin in the second year and are progressively scaffolded through to the fourth year. These blocks expose students to various fields of occupational therapy, including occupational therapy applied to physical conditions. The overall structure is as follows:

- 2nd year: Students attend clinical settings once a week for 8 weeks (no splint-making, as splint training begins in the third year).

- 3rd year: Students attend three times per week for 8 weeks and start fabricating splints.

- 4th year: Students attend clinical settings daily for 6 weeks.

Researchers emphasise the importance of applied learning for developing professional competence, suggesting that clinical exposure, rather than classroom instruction, plays a more pivotal role in student development [7,23]. This relationship has profound implications for curriculum design.

De Witt et al. (2019) [24] found that international studies emphasise the importance of such clinical or hands-on exposure in fostering both clinical competence and professional identity in OT students. These practical learning environments develop students' ability to adapt to real-world clinical complexities, improving their problem-solving and therapeutic reasoning skills. For instance, splinting, a core skill for treating various hand injuries, requires both technical proficiency and adaptability to patient-specific conditions [14]. Integrating hands-on experiences into undergraduate training is crucial, and the current focus on clinical fieldwork as part of applied learning reinforces this need. This literature suggests that improving access to resources and providing comprehensive practical opportunities could help bridge the gap between theory and practice. Ultimately, strengthening undergraduate OT training at UKZN—particularly through practical experiences such as splinting—will help produce more competent graduates who are better equipped to serve in the public health sector. This can enhance access to rehabilitation services for underserved populations in KZN province and improve the well-being of individuals affected by hand injuries in these communities. However, several challenges hinder the effective implementation of practical learning experiences, including difficulties in finding suitable clinical fieldwork placements and supervisors [25]. These have a negative impact on students' exposure to specific OT fields, such as splinting as part of

hand rehabilitation and their overall experience levels. Such placements have an impact that extends beyond the South African setting, as it is noted that a lack of supervision, limited access to clinical sites, and high academic pressure contribute to students' feelings of being underprepared and stressed internationally [24].

While the value of clinical preparedness is well documented, there is a scarcity of research that contextualises this within South African OT programs, specifically those operating in resource-constrained provinces like KZN. Furthermore, few studies have explored how undergraduate students perceive their readiness for splinting as a specific skill within hand rehabilitation. Given the high burden of hand injuries in KZN and the uneven distribution of therapists across the public and private sectors, this study is crucial for understanding how current training methods affect students' competence and long-term workforce readiness. The section below focuses on Kolb's Experiential Learning Theory in relation to this study.

### Kolb's experiential learning theory

Widely recognised and applied since its inception [23,26], Kolb's Experiential Learning comprises four stages of learning: concrete experience, reflective observation, abstract conceptualisation and active experimentation [7,23]. Despite its prominence, Morris (2020) [26] critiques it for lacking robust theoretical support and clarity in its components. Morris (2020) [26] argues that the theory should be more contextually responsive, emphasising contextual richness, context-specific and practical experiences to enhance active experimentation. In essence, Morris (2020) [26] suggests that the model should incorporate contextual considerations to be more responsive, a finding that aligns with this study's results.

This study advocates for integrated educational models that blend theory with experiential learning, as supported by researchers [23,26]. Kolb's Experiential Learning Theory [7,23] contextualises this study's findings, highlighting the limitations of curriculum structures that prioritise theory over practical training, which may hinder procedural skill consolidation. The findings underscore an urgent need for curriculum developers to integrate deliberate and structured clinical exposure into the OT program at UKZN. This need arises in the context of the unequal distribution of therapists between South Africa's private and public sectors, requiring a curriculum that equips students with adaptability and resourcefulness for diverse, often resource-constrained environments. Contextually relevant training [26] could better prepare students to serve South Africa's diverse populations. The aim and objectives of this research were achieved as outlined in the conclusion section below.

## Methodology

### Study design

A qualitative study design was employed to gain an in-depth understanding of undergraduate OT students' perceptions of their training. This approach is particularly suited for capturing students' complex and subjective perspectives, which cannot be adequately represented through quantitative methods. The Consolidated criteria for reporting qualitative research (COREQ): a 32-item checklist for interviews and focus groups was followed to ensure that all method-related items were adhered to [31].

### Population

The study was conducted at the UKZN, Westville campus. The targeted student cohort, comprising just under 70 registered combined 3rd and 4th year Bachelor of Occupational Therapy students in the College of Health Sciences, was selected. A purposive sampling strategy was used to select participants who had been exposed to splinting theory and practice, which was introduced in the 3rd year of study. This method was chosen to ensure the inclusion of students with relevant knowledge and experience related to the research topic. Purposive sampling was chosen because it enabled the intentional selection of individuals most likely to provide rich, in-depth, and context-specific data, which would not have been possible with random sampling [27].



## Inclusion and exclusion criteria

The inclusion criteria comprised 3rd and 4th year OT students from UKZN with physical fieldwork experience and exposure to hand rehabilitation, particularly splinting. First and second year students, as well as those without prior physical fieldwork experience or exposure, were excluded from this research because they had not yet covered splinting in their curriculum.

## Data collection process

The data collection process commenced following ethical clearance from the UKZN Biomedical Research Ethics Committee (BREC). Class representatives of 3rd and 4th year OT students were contacted via WhatsApp and requested to distribute an information sheet along with a Google form. This form was used to collect (prospective) participants' contact information, level of study, and sex, which enabled the researchers to identify them and obtain informed consent. The information sheet outlined the study's purpose, explained that participation was entirely voluntary, and stated that participants could withdraw at any stage without consequences or penalties. It also detailed any potential benefits of participation. It included contact information for the researchers, which allowed participants to ask questions or seek further clarification in accordance with UKZN's ethics policy [22].

Individual, once-off semi-structured interviews were conducted using the Zoom platform. Each researcher selected participants who consented. A literature-informed interview guide was used to ensure consistency. This guide was piloted with recent OT graduates and refined to improve clarity. Recruitment and data collection spanned from 22 July to 12 September 2024. Additionally, peer debriefing and member checking were used where necessary to validate interpretations and ensure the credibility of the findings. The researchers aimed for data saturation and sufficient depth of perspective to provide a meaningful understanding of participants' experiences, thereby strengthening the reliability of the study [28]. Data saturation was achieved when the researchers had collected sufficient data to draw the necessary conclusions, and further data collection yielded no additional insights of value into the study. Trustworthiness and accuracy of the data were ensured through several strategies grounded in credible qualitative research practices. Participation in the study was voluntary, and to promote anonymity and honest responses, participants were asked to refer to themselves as "Participant" during online Zoom interviews. All interviews were audio-recorded with participant consent and later transcribed verbatim (using Microsoft Word Online) to maintain the accuracy and integrity of the data. Coding was conducted by two researchers and verified by the other two researchers to enhance reliability. Intercoder agreement was assessed using a consensus, where discrepancies in coding were discussed and resolved collaboratively by all researchers to ensure consistency and rigour in theme development. The research team consisted of four student researchers, supervised by an experienced academic.

Reflexivity, which entails continuously critiquing one's own beliefs, subjectivity, and their influence on the research process [29], was maintained throughout the study to minimise bias. A transparent decision-making process and regular collaborative discussions with the research team supported this. Data triangulation among the research team further strengthened the credibility of the findings. A clear audit trail was kept, and all data were securely stored on a restricted-access online platform. All raw data files are available from the Open Science Framework (OSF). The complete reference is as follows: UKZN undergraduate research Raw Data. OSF; 2024. Available from: https://osf.io/yvd7r under file: https://osf.io/yvd7r/files/osfstorage. Contact the corresponding author for further details.

## Data analysis

A hybrid approach was employed during the thematic analysis, as it allows for both inductive and deductive methods [30]. While Braun and Clarke's six-step process, namely familiarising with the data, data coding, identifying themes, reviewing themes, defining and naming themes and producing the report [31], guided the analysis, some themes emerged



inductively from the data. Others were informed deductively through existing literature and research objectives that aligned with the study's focus, e.g., 'explore the OT students' perceptions of their readiness and preparedness'. To safeguard the inductive component, initial codes were generated directly from participants' words and independently reviewed by other researchers before being examined alongside deductive expectations, allowing unexpected insights to be retained. Four themes, each with corresponding subthemes, emerged, supported by quotes from participants.

### Ethical considerations

The study adhered to the World Medical Association's Declaration of Helsinki, ensuring informed consent, privacy, non-maleficence, beneficence, and confidentiality [32]. The UKZN Biomedical Research Ethics Committee (BREC) reviewed the research proposal and granted both gatekeeper permission and ethical clearance, ethics number BREC/007090/2024. Written and witnessed informed consent was obtained from all participants.

## Results

This section focuses on the findings related to the participants' (students') demographics and four emerging themes.

### Demographic profile of the participants

This research consisted of 16 participants: n = 5 males and n = 11 females undergraduate OT students at UKZN, of whom six were in their 3rd year, and 10 were in their 4th year. The number of physical blocks completed by the participants ranged from two to three. The number of splints made during clinical fieldwork showed that two participants made no splints, seven made between one and three splints, and seven made four or more. Refer to Table 1 for details.

### Themes

Four themes and corresponding subthemes emerged in this research, summarised in Table 2 below. The four themes are student readiness and preparedness, challenges in splinting, comparison of readiness in the 3rd year and 4th year, and

**Table 1. Demographics of Participants.**

| Participants | Sex | Year of study | Number of physical blocks | Number of splints made while on clinical fieldwork |
|---|---|---|---|---|
| 1 | Female | 4th year | 3 | 1 |
| 2 | Male | 3rd year | 2 | 3 |
| 3 | Female | 4th year | 3 | 4 |
| 4 | Female | 3rd year | 2 | 0 |
| 5 | Female | 3rd year | 2 | 2 |
| 6 | Male | 4th year | 2 | 4 or 5 |
| 7 | Female | 4th year | 2 | 3 |
| 8 | Female | 4th year | 2 | 3 |
| 9 | Male | 4th year | 3 | 4 |
| 10 | Male | 3rd year | 2 | 1 |
| 11 | Female | 4th year | 2 | 3 |
| 12 | Female | 4th year | 3 | 4 |
| 13 | Male | 3rd year | 2 | 1 |
| 14 | Female | 3rd year | 2 | 0 |
| 15 | Female | 4th year | 3 | 4 or 5 |
| 16 | Female | 4th year | 3 | 1 |

**Table 2. Themes and Subthemes.**

| Theme | Subthemes |
|---|---|
| **Theme One: Student Readiness and Preparedness** | 1.1: Perceived Preparedness for Clinical Fieldwork - *"I wouldn't say it's prepared me enough."* |
| | 1.2: Confidence Levels - *"I'm not confident enough to do a splint on my own."* |
| | 1.3: Theory Knowledge and Practical Skills - *"I have maybe the theory knowledge but not the application knowledge."* |
| | 1.4: Influence of Clinical Fieldwork Experience on Preparedness - *"The level of preparedness I feel… has only changed with experience at fieldwork."* |
| **Theme Two: Challenges in Splinting** | 2.1: Limited Exposure to Hand Rehabilitation - *"I haven't had… an opportunity to do hand therapy."* |
| | 2.2: Gaps in Key Skills and Knowledge - *"I don't have all the skills and all the knowledge…"* |
| | 2.3: Practical Limitations - *"I can't make a splint quick enough."* |
| **Theme Three: Comparison of Readiness in 3ʳᵈ Year and 4ᵗʰ Year** | 3.1: Limited Improvement in Preparedness – *"I don't think there is much of a difference."* |
| | 3.2: Theoretical Growth and Practical Competence |
| | 3.3: Confidence in Practical Skills for Clinical Fieldwork - *"Definitely not competent…"* |
| | 3.4: Motivation to Specialise in Hand Rehabilitation - *"I think I'll be interested…"* |
| | 3.5: Deterrents to Specialisation – *"I don't feel I'm confident enough to do anything with hands."* |
| **Theme Four: Suggested Solutions to Bridge the Gaps** | 4.1: Integration of Theory and Practice - *"It's very difficult to apply what we learned in anatomy and physiology to actual splinting…"* |
| | 4.2: Need for Increased Practice Time - *"Maybe we can have more time…".* |
| | 4.3: University and Clinical Support - *"The university can better support us by…"* |
| | 4.4: Adequacy of Curriculum Content - *"The curriculum shouldn't be just two to three days…"* |

suggested solutions to bridge the gaps. The themes are explored using verbatim quotes to capture and represent the participants' perspectives authentically.

### Theme one: Student readiness and preparedness

**Subtheme 1.1: Perceived preparedness for clinical *fieldwork – "I wouldn't say it's prepared me enough."*.** Most students expressed that they did not feel adequately prepared to make splints at clinical fieldwork despite receiving the necessary training in class. They attributed their lack of readiness to insufficient practical sessions and limited exposure to various types of splints. Participants expressed the following views:

*"We've had quite a few splinting sessions, but we've only actually physically made about two functional splints... After doing that and that was quite a while ago, I don't think it was really enough for us to know exactly that with confidence."* (Participant 4, 3ʳᵈ Year)

*"Due to the fact that we spend less than five days doing our pracs [splinting practical sessions in class], I feel like it's not enough for me because we need more time to like prepare ourselves in order to be able to do splinting"* (Participant 14, 3ʳᵈ Year)

*"We haven't had adequate training in terms of we're just given, I think two or three lessons in 3ʳᵈ year. And then two to three lessons, also in 4ᵗʰ year and then that was it. You learn a lot from actually, from your clinical supervisors".* (Participant 9, 4ᵗʰ Year)

*"No, I wouldn't say it's prepared me enough. I think we have learnt splinting in two or three sessions, where we're just learning the basics of them, not getting into details of how splinting is, but splinting is our own [OTs] thing. Only OTs [occupational therapists] can do splinting, so I feel like we need more lectures on splinting because with the knowledge I have right now, I don't think I am fully ready for splinting people with different splints and different diagnoses."* (Participant 6, 4th Year)

*"No, I feel like we needed more time [for] splinting. Doing two splints, 'out of the how many splints that are available?', and they were the easiest splints, so it hasn't really equipped me enough to go to the hospital and be like, 'okay, let's make a splint for this patient.'"* (Participant 5, 3rd Year)

**Subtheme 1.2: Confidence levels – *"I'm not confident enough to do a splint on my own."*.**  Students reported low confidence in their ability to perform splinting tasks independently. Although they noted a slight improvement in confidence after exposure to clinical fieldwork, they generally felt that their confidence levels remained suboptimal. This perceived lack of confidence was attributed to insufficient training in both practical and theoretical aspects of splinting. Participants had the following to say:

*"I got like the exposure I got on fieldwork and not at school [university], at field work, while I was doing my physical prac [clinical fieldwork], right now it's kind of really helped me boost my confidence in making the splints"* (Participant 11, 4th Year)

*"I think I do have some of the skills, but not all of them. As much as I have practiced making a splint on prac sites, I've only been able to make one splint in fieldwork, and I was not exposed to making other splints for different clients with different diagnoses."* (Participant 10, 3rd Year)

*"No, not at all. I'm sorry to say, I don't feel it with the time spent on splinting in both third year and in fourth year. I'm not confident enough to do a splint on my own."* (Participant 7, 4th Year)

*"In my opinion, no, because I still feel that I'm not confident in making a splint on someone with a real injury, as we've only practised on fellow students without injuries."* (Participant 8, 4th Year)

**Subtheme 1.3: Theory, knowledge and practical skills - *"I have maybe the theory knowledge but not the application knowledge."*.**  The participants felt that their theoretical knowledge did not translate well into practical skills. They mentioned that, while they understood hand anatomy and splinting concepts, their limited hands-on experience hindered their ability to apply theoretical knowledge effectively in real-world clinical settings. Below are some supporting quotes from participants.

*"It was more with the splinting that I felt that I wasn't very competent because I wouldn't know exactly which splint to issue for the patient"* (Participant 3, 4th Year)

*"Well, I feel maybe we spend more time learning about it in theory and less time actually making the splint."* (Participant 7, 4th Year)

*"I have maybe the theory knowledge but not the application knowledge"* (Participant 8, 4th Year)

**Subtheme 1.4: Influence of clinical fieldwork experience on preparedness – *"The level of preparedness I feel… has only changed with experience at fieldwork."*.**  Students attributed a portion of their increased preparedness and confidence to the hands-on experience gained during clinical fieldwork, rather than to improvements in the curriculum between 3rd and 4th years. They specifically credited the guidance of clinical

supervisors and real-world practice as key factors for their growth and development. Moreover, the students reported that the university's curriculum still fell short of providing sufficient practical training opportunities. Participants shared the following insights:

*"The placement I'm in has made me see that it has really, really, really prepared me or has taught me so much more about splinting and conditions… but the module itself doesn't necessarily give you that confidence to go out and make splints on your own." (Participant 16, 4th Year)*

*"The exposure I got on fieldwork, and not at school [university], at fieldwork while I was doing my physical prac, right now it's kind of really helped me boost my confidence in making the splints." (Participant 11, 4th Year)*

*"The level of preparedness I feel like has only changed with experience at fieldwork. So, seeing a bigger variety of the different splints that can be made and how they can be used in practice." (Participant 8, 4th Year)*

**Theme two: Challenges in splinting in hand rehabilitation at UKZN**

**Subtheme 2.1: Limited exposure to hand rehabilitation -** *"I haven't had… an opportunity to do hand therapy."*. Responses showed that students had varying degrees of exposure to hand rehabilitation cases during clinical fieldwork, with some reporting no exposure at all. They reported that this discrepancy had an impact on their ability to practice and refine splinting skills in a real-world setting. Participants provided the following feedback:

*"I haven't had like an opportunity to do hand therapy because I haven't had a patient with hand deficits." (Participant 2, 3rd Year)*

*"No, I didn't work with a patient who needed hand rehabilitation, I only did exercises." (Participant 14, 3rd Year)*

*"Not my patients, but I helped the OTs [occupational therapists] at the hospital by assisting with hand rehabilitation and splinting. So not by myself." (Participant 5, 3rd Year)*

**Subtheme 2.2: Gaps in key skills and knowledge –** *"I don't have all the skills and all the knowledge…"*. The students expressed concerns about deficiencies in their skill set and knowledge base, particularly regarding specific splinting techniques and the appropriate selection of splints based on different hand conditions. They emphasised the need for targeted training in clinical decision-making and adherence to protocol splinting. These are the students' thoughts and responses:

*"But for me, I think only one patient needed splints. There was a patient that I did get… his hands were in very bad condition, but it had been too long since the time that he had his condition or incident for there to be any progress for splinting now. So, it would have been useful to have known about this." (Participant 4, 3rd year)*

*"Ohh, I lack... I'm always doing tight splints." (Participant 14, 3rd Year)*

*"I think because I haven't gotten much exposure to hand rehabilitation. I haven't been exposed to all the different hand conditions so that I can apply all the skills and all the knowledge that I have." (Participant 12, 4th year).*

*"We are not sure if it is the correct way, and if you are looking to splint, it requires more creativity, things like that. You need to know your theory plus your practical because you are already endangering your client." (Participant 1, 4th year)*

**Subtheme 2.3: Practical limitations –** *"I can't make a splint quick enough"*. Some students expressed frustration with the time-consuming nature of splinting real patients, attributing this to their limited practice and familiarity with the



task. They expressed that their lack of experience with splinting resulted in a slower completion time, which in turn eroded their confidence in performing splinting tasks independently. Participants said the following:

*"That's the one thing that I've battled with when I've been making splints in my fieldwork: I can't make a splint quick enough, and then the patient ends up sitting for quite a while, while I make the splint." (Participant 8, 4th Year)*

**Theme three: Comparison of readiness in 3rd year and 4th year**

**Subtheme 3.1: Limited improvement in preparedness – *"I don't think there is much of a difference."*.** Participants noted that although there was a slight increase in confidence and knowledge from the 3rd to the 4th year, they still felt inadequately prepared for the challenges of clinical fieldwork. Participants explained:

*"Not so much. In third year, I knew two splints like two static splints. And this year, I've added like two dynamic splints… but compared to clinical fieldwork as a whole, I don't think there is much of a difference." (Participant 1, 4th Year)*

*"It's definitely the same. It is definitely the same because we only go through about two lectures on hand rehab [rehabilitation]. And that's the end of it. Then you go into the splinting where they actually show you how to do it… So, it's the same. It's the same. I don't feel like it was a step up from what I knew last year." (Participant 3, 4th Year)*

*"I feel it's pretty much the same. I don't feel like I've gotten more prepared in any way." (Participant 6, 4th Year)*

**Subtheme 3.2: Theoretical growth and practical competence.** Students in both 3rd and 4th years reported enhanced theoretical comprehension of hand rehabilitation and splinting techniques. However, the theoretical growth failed to translate into corresponding practical skills, as 3rd year students had limited exposure to splint-making, and 4th year students still felt underprepared for clinical applications. They reported that this disparity left them feeling inadequately equipped to manage complex cases independently. They had the following to say:

*"I think I'm more confident in hands or splinting now compared to 3rd year, it is confidence from what I've seen, what I've done in the field… In 3rd year, I got to learn about splints but never got to do a splint in a clinic and clinical fieldwork." (Participant 12, 4th Year)*

*"From 3rd year, maybe, I would say three out of ten, but from 4th year, I'd say six out of ten. Being exposed more while I was on prac with splinting material and making the splints really helped improve my confidence." (Participant 15, 4th Year)*

*"We've had quite a few splinting sessions, but we've only actually physically made about two, like functional splints, but they haven't been dynamic splints. They've just been resting splints. And we also made one other or two other small type splints, which are a knuckle duster and one for writing splint, and there's also a lot more variance of splints, which I feel like we would have needed more practice on doing." (Participant 4, 3rd Year)*

*"I think I have some of the skills, but not all of them... I've only been able to make one splint in fieldwork, and I was not exposed to making other splints in fieldwork and for other clients with different diagnoses." (Participant 10, 3rd Year)*

*"Definitely the theory, I think I do have a bit of it. Not 100%, maybe 60%... I don't have all the skills and knowledge because I haven't gotten much exposure to hand rehabilitation." (Participant 12, 4th Year)*

*"I possess some of them [key skills and knowledge areas]. I feel like I constantly have to go back to my anatomy... But I don't feel that much confidence practising without supervision, especially when it comes to hands." (Participant 9, 4th Year)*



**Subtheme 3.3: Confidence in practical skills for clinical fieldwork - *"Definitely not competent…"*.** Third-year students expressed self-doubt about their ability to perform splinting independently in the clinical fieldwork setting. They emphasised the need for additional practice and supervision, acknowledging that they would require assistance in managing real patients with hand injuries. They also note the potential risk of incorrect splinting, which exacerbated their uncertainty and reinforced their perceived incompetence. Fourth-year students similarly reported a lack of confidence in their splinting skills, with many admitting that they would still struggle to handle hand rehabilitation cases without supervision. They felt that their limited clinical exposure had left them uncertain of their abilities in a clinical fieldwork setting. Participants explained:

*"After doing that [university training]... I don't think it was enough for us to know exactly, with confidence, that we can now do it all on our own. We would probably need some assistance." (Participant 4, 3rd Year)*

*"It's very dangerous to put a person in a splint they're not prepared for... it can exacerbate the problem. it's something very sensitive, but if you know exactly what you're doing, it's really effective." (Participant 4, 3rd Year)*

*"Definitely not competent, I must be honest. It's different in a clinical setting than from a theory perspective." (Participant 3, 4th Year)*

*"Yeah, I do have some of the skills, just like the confidence is lacking... it's just that we don't get enough time to practice making splints." (Participant 11, 4th Year)*

**Subtheme 3.4: Motivation to specialise in hand rehabilitation - *"I think I'll be interested…"*.** A few students expressed an interest in specialising in hand rehabilitation, motivated by their experiences in the field. This enthusiasm was notable, particularly among 3rd year students, who had relatively limited exposure to splinting within hand rehabilitation. They recognised the value of developing expertise in this area to address the existing gap in hand rehabilitation services within the community. Their comments were as follows:

*"I think I'll be interested… as an OT [occupational therapy student], everything is about function, and what makes us [people] able to function in most areas of occupation is our hands." (Participant 1, 4th Year)*

*"Hand rehabilitation is a great thing you get the ability to learn, to do your creativity in terms of making splints, designing different types of splints and so on and doing splints for different types of people with different conditions." (Participant 10, 3rd Year)*

*"I think I would find… it interesting in participating… specialising in hand, although I doubt it, it would be challenging." (Participant 2, 3rd Year)*

*"Yes, I would. I do plan on doing a postgraduate diploma in hand therapy." (Participant 7, 4th Year).*

*"When I was doing my physical thing [practical], when I was seeing my supervisors, they did the course hands, in hand rehabilitation and whatnot. And then I kind of enjoyed it. So, I think I might be interested in it." (Participant 11, 4th year)*

**Subtheme 3.5: Deterrents to specialisation – *"I don't feel I'm confident enough to do anything with hands"*.** Conversely, some students were discouraged from pursuing a specialisation in hand rehabilitation due to their self-perceived deficiencies in splinting skills and confidence. They felt insufficiently prepared to manage complex hand cases independently, which deterred them from considering specialising. Participants said the following:

*"No, I think I don't have an interest in it… It's okay in a hospital setting, but I wouldn't pursue it as a postgrad specialisation because I don't feel I know enough about it." (Participant 3, 4th Year)*

*"It's not a bad field to pursue, but I'm not comfortable with hand rehab and splinting because I don't feel I have the skills needed, so I would rather focus on other areas." (Participant 15, 4th Year)*

*"No, currently no. I don't feel I'm confident enough to do anything with hands, so going completely into hand rehab, definitely not." (Participant 8, 4th Year)*

**Theme four: Suggested solutions to bridge the gaps**

**Subtheme 4.1: Integration of theory and practice.** Participants reported that a more seamless integration of theoretical modules and practical sessions would enhance their preparedness for clinical fieldwork. Participants explained:

*"I think those modules have helped us a lot in terms of understanding the basic anatomy of the hand. And because we require, we need to know which muscles are affected in order for us to know the type of splints that we are going to make and to see and to also understand which muscles actually are affected. This is mostly needed when making splints." (Participant 10, 3rd year)*

*"I think more time can be spent on looking at the different splints in detail. Rather than only focusing on the four splints that we practically make as part of our training." (Participant 8, 4th Year)*

*"Well, I feel maybe we spend more time learning about it in theory and less time actually making the splint." (Participant 7, 4th Year)*

**Subtheme 4.2: There is a need for more practice time - *"Maybe we can have more time…".*** Participants suggested that the university should allocate more time for practical splinting training, recommending that sessions be extended and training commence earlier in the program, ideally from the second year onwards. They believed this would enable the progressive development of foundational skills. Participants suggested the following:

*"Maybe we can have more time… like the curriculum shouldn't be like two to three days, we can have a week... The university can implement a curriculum that helps us cover everything that we might face in fieldwork." (Participant 1, 4th Year)*

*"I think the university must try to expose students to more cases that require students to make splints because we might have learned how to do the splints, but not all of us have been exposed to making the splints on the fieldwork sites cause, some of us were maybe treating patients with no deformities in the hand." (Participant 10, 3rd Year)*

*"Maybe we get more chances and more practical work on splints that you can master, or can try mastering the hand rehabilitation." (Participant 6, 4th Year)*

**Subtheme 4.3: University and clinical support.** Students acknowledged the valuable contribution of clinical supervisors to their learning experience during clinical fieldwork. However, they indicated that the university could provide more comprehensive support to facilitate a smoother transition from theory to practice. Specifically, they recommended increasing supervised practice time before clinical placements to better prepare them for fieldwork placements. Their feedback was as follows:

*"They should allow for students to become more comfortable with the material required to make splints, like the thermoplastic, the heat gun and all of that, and also practice making more different splints as well." (Participant 7, 4th Year)*

*"You learn a lot from actually from your clinical supervisors... just from your clinical supervisors, more than the university." (Participant 9, 4th Year)*

*"I think the university must try to expose students to more cases that require students to make splints."* (Participant 10, 3rd year)

**Subtheme 4.4: Adequacy of curriculum content - *"The curriculum shouldn't be just two to three days…".*** Students perceived the splinting section of the curriculum to provide foundational yet incomplete training. They emphasised the need for more extensive coverage of key topics, including anatomy, physiology, and diverse splinting practices, to ensure a more comprehensive understanding. Furthermore, they advocated for additional practice time and lectures to reinforce their knowledge and practical skills in hand rehabilitation. These are the participants' thoughts and responses:

*"The curriculum shouldn't be just two to three days… I know that we can't cover all the splints, but the most basic and common ones we find in hospitals should be taught thoroughly."* (Participant 1, 4th Year)

*"I think they have played a very crucial role in terms of making us easily understand, like the intervention we give… in making splints, you will know what kind of splint you make, according to the injuries in terms of like the hand part"* (Participant 2, 3rd Year)

## Discussion

The discussion draws on the theoretical foundation of this study and its two primary objectives to unpack the findings. Specifically, Kolb's Experiential Learning Theory, student readiness and preparedness, and proposed solutions for enhancing the OT UKZN program's hand rehabilitation component are highlighted below. Although not explicit in their accounts, the findings indicate movement through only part of Kolb's cycle: students engage in reflective observation and abstract conceptualisation through theory-based instruction but have limited opportunities for concrete experience and structured active experimentation. This framing sets the foundation for the discussion that follows.

### Student readiness and preparedness

The findings from this study highlight a notable discrepancy between students' theoretical knowledge and their perceived readiness to apply splinting techniques in clinical settings. Despite increased exposure to theoretical content in the 3rd and 4th years, students consistently reported feeling underprepared, uncertain, and low in confidence in their splinting abilities. These concerns are similar to those raised by Ghela et al. (2022) [8], who identified a critical shortage of OTs specialising in hand rehabilitation in KZN, thereby limiting opportunities for supervised clinical practice in this area. This pattern suggests that both exposure and its sequencing influence confidence. This study highlights the need to balance classroom teaching with hands-on [23,33] clinical fieldwork to enhance students' readiness for splinting techniques in OT training, thereby reducing the risk of patients receiving incompatible splints that could endanger their health. Students value classroom learning for its structured, often ungraded setting, which builds confidence. However, limited access to practical experience in hand rehabilitation hinders their skill development, as shown by this study and supported by Ghela et al. [8]. These findings are supported by Kolb's Experiential Learning Theory [7].

The above findings support and expand upon available global [19] and South African [20] research, which suggests a growing demand for therapists skilled in managing traumatic hand injuries. Students' confidence appeared to fluctuate between teaching and clinical periods, suggesting that irregular exposure may disrupt continuity in their learning. This highlights the value of practice that is revisited and reinforced over time rather than delivered in isolated blocks. The students' anxiety around preparedness may affect workforce development, as some may be discouraged from specialising in hand rehabilitation. This concern is supported by Ned et al. (2020) [9], who reported that most experienced hand therapists work in the private sector, worsening access to care in the public health system.

Interestingly, despite the challenges, a few students showed interest in specialising in hand rehabilitation. This may suggest that their motivation offsets perceived skill deficits, aligning with self-efficacy theory, which emphasises the role of confidence in forming one's identity [34,35]. However, the overall trend of low confidence signals a disconnect between curriculum structure and clinical readiness, warranting urgent curricular reform. Emphasis on experiential learning could assist in this regard [7]. Consistent with the literature [22,23], this study reinforces the value of experiential learning in closing the gap between theory and practice. Although students acquire substantial theoretical knowledge, their ability to apply it in practice is hindered by limited hands-on opportunities. Kolb's Experiential Learning Theory supports this, asserting that concrete experience and active experimentation are crucial for skill development [23].

This study also reveals inconsistencies in clinical training. While some students had meaningful hands-on exposure, others were confined to observational roles. These inconsistencies likely reflect broader systemic issues such as placement shortages, limited supervisory capacity, and resource constraints, as noted by Naidoo (2013) [25]. These factors may be beyond the control of UKZN and the clinical fieldwork sites hosting their students. These variations risk producing uneven graduate competence and, ultimately, affecting the quality of care delivered. Students with greater hands-on clinical exposure in hand rehabilitation gained a significant learning advantage, as supported by Wijnen (2022) [23], Morris (2020) [26] and Kolb (1984) [7]. However, those restricted to observational roles in the OT program's hand rehabilitation section reported frustration with repetitive, limited training experiences, mirroring the systemic placement and supervision constraints identified by Naidoo (2013) [25] and evidenced in this study.

## Student-proposed solutions for enhancing splinting training

Students' recommendations to increase practical training time, introduce splinting earlier in the curriculum, and diversify clinical placements directly respond to the challenges identified in both this study and the literature. These recommendations align with Van Stormbroek and Buchanan (2018) [22], who advocate for curricula that reflect common clinical presentations and emphasise iterative skill practice. Moreover, these findings draw on Kolb's Experiential Learning Theory, which spans concrete to active experimentation [7,26]. Specifically, students could transition from learning splinting patterns in class to applying abstract problem-solving skills to fabricate suitable splints tailored to diverse patient conditions or injuries. The proposal to introduce splinting earlier aligns well with scaffolding learning principles [36], which advocate introducing foundational skills early and building on them systematically to foster competence and confidence. These patterns indicate that confidence develops gradually and is strengthened through repeated reinforcement, offering a broader understanding of why students consistently request more opportunities to practise. For instance, introducing basic splinting techniques and materials at a junior level could enhance students' preparedness. By providing more time to practice and refine skills, this scaffolding approach could mitigate the confidence deficit observed in clinical practice. Furthermore, students' calls for diversified fieldwork placements address healthcare access disparities and uneven therapist distribution, as noted by Ghela et al. (2022) [8]. Expanding clinical placement opportunities, particularly in underserved areas, can enrich student experiences while addressing workforce inequities. However, achieving these outcomes requires enhanced institutional partnerships with the public health sector, improved supervisory capacity, and strategic curriculum reform, which may take some time. In the interim, minor adjustments, such as those outlined above, can be implemented immediately to enhance learning in line with Kolb's Experiential Learning Theory.

In closing, the above discussion suggests that students in this research move through only part of Kolb's Experiential Learning Cycle, gaining reflective observation and abstract conceptualisation through theory-based teaching, but with sparse concrete experience and structured, consistent opportunities to foster active experimentation. This incomplete movement through the learning cycle may explain why additional exposure on its own does not necessarily build confidence or a sense of readiness.



## Conclusion

Despite a foundational theoretical understanding of splinting in hand rehabilitation, this study's findings highlight significant concerns about OT students' readiness and preparedness for splinting in clinical fieldwork. Although they acquired theoretical knowledge, students felt underprepared and uncertain due to the perceived insufficiency of practical training and limited clinical exposure. This lack of confidence and competence may deter students from pursuing a specialisation in hand rehabilitation, exacerbating the existing shortage of qualified therapists in this area and compromising patient care. The study's findings underscore the importance of balancing theory and practice in OT education by ensuring students are supported and move through all stages of Kolb's Experiential Learning Cycle.

To address these gaps, students recommended strategies such as increasing practical time dedicated to splinting, introducing training earlier in the program, expanding the splinting curriculum to cover a broader range of splint types, and strengthening university support structures. These recommendations resonate with existing literature, which emphasises the crucial role of hands-on experience in fostering confidence and competence. By implementing these changes, the university can better prepare students for clinical practice, enhance their overall competency in splinting as a part of hand rehabilitation, and ultimately improve the quality of hand rehabilitation services in South Africa.

Addressing the identified gaps requires engaging stakeholders, such as university training staff, to implement feasible changes within their mandates through appropriate channels. For example, a complete curriculum may require removing certain elements to allocate additional time. Alternatively, realigning the curriculum could facilitate a smoother transition from junior to senior years, boosting students' confidence in hand rehabilitation. Expanding and diversifying clinical fieldwork placements, particularly in underserved rural areas, would provide vital real-world experience and exposure to a range of hand conditions. For greater clinical impact or if other institutions face similar challenges, the OT profession's national body could reassess research priorities and teaching methods to ensure adequate exposure, ultimately benefiting these institutions and the communities they serve by addressing the shortage of OTs in this critical field. These changes would help ensure that students progress through the whole cycle, from foundational theoretical understanding to meaningful hands-on experience and active experimentation. See the recommendations and impact on practice section below for a concise overview of this study's contributions, which ensure that students meet competency benchmarks throughout their training.

## Recommendations and impact on practice

To achieve a positive impact, targeted educational reforms can better prepare graduates for clinical practice and strengthen the OT workforce in hand rehabilitation. These reforms could be enhanced through a revised policy perspective and collaboration between educational institutions and healthcare providers. The following recommendations aim to address gaps in this field to enhance OT students' readiness for clinical practice:

i. Enhance practical training: Although intense in theory, the current OT hand rehabilitation section at UKZN should include additional practical training to bridge the gap between classroom learning and clinical fieldwork. Specifically, splinting should be introduced earlier in the curriculum, with extended classroom practice sessions that gradually build skills, starting with basic tasks and advancing to more complex practice to help students develop competence progressively.

ii. Introduce a logbook: The university should introduce a logbook to track students' practical skills and document the conditions they have been exposed to, as well as splints, pressure garments, and other assistive devices fabricated over the years of clinical fieldwork. This logbook should be monitored and evaluated regularly, such as twice during each clinical fieldwork block (at mid-term and final block evaluations), to ensure students meet the required competencies.

iii. Diversify clinical fieldwork placements: Given the challenges of finding suitable placements, where practical, the university should assist students in accessing a wider range of placements where hand rehabilitation is practiced, including in rural or underserved areas. This will expose students to various splinting techniques and hand injuries, increasing their readiness to manage clients independently. Students should be encouraged to pursue a hand rehabilitation clinical placement during an elective (ungraded or unmarked) block to enhance exposure.

iv. Implement mentorship programs: The university should consider establishing mentorship programs that pair students with experienced practitioners in hand rehabilitation. This could also involve structured supervision during fieldwork placements to ensure that students receive adequate support as they develop their practical skills in hand rehabilitation.

v. Conduct further studies: Future research should explore perceptions of splinting readiness across multiple institutions with more diverse samples. Longitudinal studies tracking graduates' competence and specialisation in hand rehabilitation would provide deeper insight into the long-term impact of undergraduate training and clinical placements. Additionally, interventional research examining the effects of enhanced practical curricula and mentorship models could provide valuable evidence to inform education policy and curriculum reform, particularly in promoting career specialisation in hand rehabilitation.

## Strengths and limitations

### Strength

This research contributes to the knowledge base of a previously underexplored area. The research employed a rigorous methodology, and ultimately, the findings can inform student training, yield better-prepared graduates and contribute to enhanced patient care, improved patient outcomes, and an overall better quality of life. Moreover, although the study focused on a single institution in South Africa, the findings hold relevance for institutions worldwide. These results could be shared at international platforms such as the World Federation of Occupational Therapists conference, potentially sparking discussions on standardised approaches that would benefit training globally.

### Limitations

This study encountered several limitations that could impact the depth of the outcomes and generalisability of the findings. The limitations are summed up as follows:

• One significant limitation was the inability to conduct focus groups due to scheduling conflicts, which limited the depth of data that could have been gathered through group discussions.

• Furthermore, a low response rate resulted in a small sample size, a characteristic of qualitative research.

• There are limited international studies that focus on OT students' preparedness in providing intervention, particularly hand rehabilitation.

• The institution-specific sample may not reflect hand rehabilitation practices or challenges faced by other universities in South Africa and internationally

• To address the limitations of this study, future studies should consider flexible scheduling or virtual platforms to facilitate focus groups, thereby enriching the depth of the data. Increasing the response rate can be achieved through follow-up reminders, simplifying data collection tools, and emphasising the value of participation. Expanding the sample size by including students from different year levels and even alumni can improve representation. Broader literature searches and collaboration with other institutions can provide more comprehensive and generalisable findings.



## Acknowledgments

We would like to thank all the students who participated in this study and generously offered their time and insights into the challenges and experiences in hand rehabilitation training.

## Author contributions

**Conceptualization:** Sizophila Gwala, Nompumelelo Mtshali, Sithembile Nene, Casey Stanley, Luther Lebogang Monareng.

**Writing – original draft:** Sizophila Gwala, Nompumelelo Mtshali, Sithembile Nene, Casey Stanley.

**Writing – review & editing:** Luther Lebogang Monareng.

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
