## [Decision Letter · Decision Letter 0]

17 Apr 2025

Dear Dr. Monareng,

Thank you for submitting your manuscript to PLOS ONE. After careful consideration, we feel that it has merit but does not fully meet PLOS ONE’s publication criteria as it currently stands. Therefore, we invite you to submit a revised version of the manuscript that addresses the points raised during the review process.

We look forward to receiving your revised manuscript.

Kind regards,

Ahmed Abdelwahab Ibrahim El-Sayed

Academic Editor

PLOS ONE

Journal Requirements:

3. In the online submission form, you indicated that “Requests for access to the data can be made by contacting the corresponding author.”.

Additional Editor Comments:

Dear authors,

Thank you for your submission to PLOS ONE. We appreciate the time and effort you have invested in this work. Following my detailed review and external reviewers' opinions, I have reached a decision that the manuscript requires major revisions before it can be reconsidered for publication.

While the topic addresses an important area, the manuscript in its current form requires significant enhancement in several critical aspects.

- The manuscript currently lacks a clear articulation of its original contribution. You are strongly advised to generate a compelling research gap, establish the significance of the study, and make a strong case for why this research is needed in your specific context. Explain how your work adds value beyond what is already well established in existing literature.

- Please elaborate on why this study is relevant within your specific population or setting. Contextual justification is crucial to demonstrate the study’s relevance and applicability.

- The sampling strategy is unclear and requires further clarification. Describe how participants were selected and justify the use of the sampling method. Additionally, please explain how you ensured the trustworthiness and accuracy of self-reported data, especially considering potential biases.

- The discussion does not meet the standards of effective academic debate. It should critically compare your findings with existing literature, reflect on contradictions and confirmations, explore theoretical and practical implications, and address limitations and directions for future research.

- As currently presented, the conclusion is not well-supported by the findings, and no actionable implications for practice, policy, or research are clearly stated. Please revise to include concrete takeaways that readers can apply or build upon.

- The manuscript contains numerous awkward and unclear statements, along with grammatical and stylistic issues that hinder readability. We advise extensive English language editing to improve clarity, flow, and academic tone. Please ensure that your writing adheres to scholarly conventions throughout.

- The reviewer panel has provided detailed and constructive feedback to help guide your revisions. We urge you to carefully review and respond to each comment, making targeted, thoughtful amendments throughout your manuscript. This will significantly improve the quality and clarity of your work and enhance its potential for publication.

Best regards

Reviewers' comments:

Reviewer's Responses to Questions

**Comments to the Author**

1. Is the manuscript technically sound, and do the data support the conclusions?

Reviewer #1: Yes

Reviewer #2: No

Reviewer #3: Yes

Reviewer #4: Yes

2. Has the statistical analysis been performed appropriately and rigorously?

Reviewer #1: No

Reviewer #2: No

Reviewer #3: N/A

Reviewer #4: Yes

3. Have the authors made all data underlying the findings in their manuscript fully available?

Reviewer #1: No

Reviewer #2: No

Reviewer #3: Yes

Reviewer #4: Yes

4. Is the manuscript presented in an intelligible fashion and written in standard English?

Reviewer #1: Yes

Reviewer #2: No

Reviewer #3: Yes

Reviewer #4: Yes

Reviewer #1: Thank you and congratulations for the interesting idea, but I must say that unfortunately the sample size is very small. Also, for such a decision, the study should be conducted simultaneously in several centers in different locations.

Reviewer #2: This manuscript requires a significant amount of improvement in:

Follow the Consolidated criteria for reporting qualitative studies (COREQ) chick list

Introduction

What is the novelty of the current study

What is known about the topic? (Background)

What is not known? (The research problem)

Why the study was done? (Justification)

Methods

How the study was designed?

How the study was carried out?

How the data were analyzed?

What is the statistical design used in the data analysis

Results:

Results need to provide answers to the questions raised/researchable problem

Results need to follow ABC (accuracy, brevity, clarity)

Kindly frame it along the following elements of results

Text to tell the story

Tables to summarize the evidence

Figures to highlight the main findings

Discussion:

The discussion section needs to be described scientifically. Kindly frame it along the following lines:

• Main findings of the present study

• Comparison with other studies

• Implication and explanation of findings

• Strengths and limitations

• Conclusion, recommendation and future direction.

• Describe sources of potential bias and imprecision.

• Generalisability of the trial findings need to be put.

Reviewer #3: Your study addresses a critical and underexplored area in occupational therapy training—student readiness for splinting in hand rehabilitation. The context of KwaZulu-Natal and the broader implications for health equity add value. However, to align with PLOS ONE’s standards for qualitative research, the manuscript requires additional methodological clarity, a deeper engagement with global literature, and improvements in structure and presentation.

Abstract: Be more specific in the abstract regarding the nature of student unpreparedness (e.g., lack of clinical opportunities, limited supervision). Add a sentence to highlight potential curriculum implications or policy relevance.

Introduction:

- Reduce redundancy (e.g., avoid repeating “splinting is a crucial technique” multiple times).

- Sharpen the focus on the specific knowledge gap your study addresses—how student confidence and competence in splinting is affected by current training models.

- Include more international perspectives to situate your study within broader educational trends.

Literature Review:

- Expand the review to include international studies on occupational therapy students’ preparedness or clinical confidence.

- Critically synthesize the literature to better support the need for this study. Consider models of experiential or competency-based education.

Metodology:

- Explain how saturation was determined and who conducted the data coding. Was intercoder agreement assessed?

- Clarify whether steps were taken to mitigate interviewer or analysis bias.

- Justify the sample size and explain participant variation in more detail.

Results:

- Present clearer distinctions between 3rd and 4th year responses if relevant to the themes.

- Consider a visual table mapping quotes to subthemes to enhance clarity and transparency.

Discussion:

- Engage with educational theory (e.g., Kolb’s Experiential Learning Cycle or Bandura’s self-efficacy theory) to better contextualize findings.

- Discuss implications for curriculum design more deeply—how might findings inform changes in timing, delivery, or assessment of splinting training?

- Expand discussion on the limitations, especially regarding generalizability beyond UKZN.

Reviewer #4: Review of article entitled: Undergraduate occupational therapy students’ perceptions of their preparedness for splinting in hand rehabilitation: an exploratory study at the University of KwaZulu-Natal- PONE-D-25-01929

General:

This worthwhile research identifies an area of occupational therapy (OT) undergraduate (UG) practical learning that is of importance but inadequately addressed in the institutional curriculum where the research was conducted. Reference is also made to the paucity of information in available literature on why students feel inadequately prepared for the use of splinting in clinical practice. Socio-economic barriers to the use of splinting are raised. The concern that poor splinting technique will deter graduates from specialising in hand rehabilitation is articulated. All these factors attest to the justification for the carrying out of this research.

The use of qualitative research methodology is appropriate with attention being paid to trustworthiness measures, ethical considerations and thematic analysis. However, some more detail related to aspects of the data collection needs to be given. The collected data in the form of quotations is of good quality and rich in useful information which suggests the open-ended questions were well designed and delivered.

The interpretation of results and conclusions offer suggestions and solutions to mitigate the curricular problems with the current practical teaching of splinting.

Abstract:

Well written, succinct and includes all the main points relevant to each section of the study.

Background:

Overall good organisation of information and relevant content.

Minor corrections

Page 5, Line 67: Is the use of the word “occupational” necessary?

Page 5, line 68: As the wording occupational therapy is used a lot perhaps consider introducing the abbreviation OT here as in occupational therapy (OT).

Page 5, lines 77-78: This point needs a citation.

Page 5, lines 79-80: This sentence is a little confusing. Perhaps consider using the word modality.as in:” which serves as a modality that is a crucial therapeutic….”

Page 5, line 83: There are instances in the manuscript where references and wording are repeated as here. Please check for similar problems elsewhere in the document and correct them.

Research question:

Minor corrections

Page 6, line 96: Suggest the addition of OT to students (to qualify who the students are) if using the abbreviation or occupational therapy students if writing in full.

This aligns with terminology to describe students in the Aim and objectives.

Aim:

Minor correction

Page 6, line 100: Could abbreviate occupational therapy to OT here.

Literature review:

Minor corrections

Can apply the abbreviation OT where appropriate.

Page 7, lines 113-114: this sentence raises the questions- How will healthcare access be enhanced? Which population’s well-being will be improved? Please clarify.

Page 7, lines 131-133; This sentence is confusing. Do occupational therapists move to the private sector due to lack of resources like splinting materials?

Page 8, line135: I would query the use of “apparent “here because the rest of the sentence suggests a definite shortage of occupational therapists.

Page 8, lines 138-140: “…the uneven distribution of occupational therapists is due to the limited number…” The addition of this wording helps with clarity. I would also suggest dividing the original long sentence into two starting the second one “ This makes it difficult….

Page 8, line148: humans

Page 8, line 151: Introduce the abbreviation multidisciplinary team (MDT) in this line as the abbreviation is used in line 153.

Page 9, line 162: I am interested to know how psychology is involved in restoring the function of the hand…

Page 9, line 183: Abbreviation UKZN already introduced.

Page 11, line 215: Abbreviate to MD.

Page 11, line 218: “emphasises”

Methodology:

What was the initial question used in the semi-structured interview? Who conducted the interviews? Were different researchers involved? Were meetings held during data collection to make sure all researchers were following the same questioning technique and adapting to changes in questioning brought about from insights from the data collection process? Were any incentives offered to participants such as free data or vouchers etc?

Was an inductive method used to analyse data?

Was verbatim transcription of interviews carried out timeously? Who did this? Was it done in Microsoft Word?

Who undertook the coding of the data? Was this done independently by researchers assigned to the task?

Minor corrections

Page 12, line 243: “was followed…. were adhered to.”

Page 12, line 257. Was approval for carrying out the research also obtained from the Head of the Occupational Therapy Department and the Dean of Student Affairs? This information could be added here or in the Ethical considerations section on Page 14.

Pages 12 and 13, lines 259-261: How and where was this data stored? Was the data anonymised? Did the information sheet give the rights of participants to confidentiality and the right to stop participation at any time with no consequence? Perhaps add to line 267-268.

Page 13, line 262: give reference for UKZN ethical policy.

Page 13, line 277; I would suggest removing wording “refer to title section for their credentials.”

Page 13, line 281: Perhaps add anonymised data to negate the chance of the reader thinking that there is any chance that identifiers would allow participants to be identified.

Results:

Rich data were collected and this allowed for pertinent and comprehensive quotations to be realised. This section is well laid out and quotations are appropriately used.

Page 16, Table 4.2; under Theme one, Subtheme 1.3, Theme three, Subtheme 3.2

I suggest that one of the quotes in the text under these Subthemes be moved from the text and used in the table.

Discussion

Although there is no specified word limit for this journal, I would suggest that perhaps in some instances fewer quotations could be used under the themes/subthemes. Some quotations basically say the same thing so the most comprehensive could be chosen. The use of a 3rd year and a 4th year quotation is good where possible to highlight differences in perceptions on a theme by students in different years of study.

Page 31, line 648: It is not preferred to begin a sentence with an abbreviation… in this case UKZN.

In general, the main points relating to undergraduate occupational therapy students’ perceptions of the preparedness for splinting in hand rehabilitation are dealt with and discussed in the discussion.

Sometimes in qualitative research “outliers” are identified in the data collection.

Perhaps consider adding a short paragraph on the dangers associated with inadequate splint making instruction.

For example: Could the lack of practice and competency in splint making and application lead to splints that are too “tight”? Could that lead to further problems for a patient? Or the fact that students recognised that the lack of adequate practice could lead to “endangering your client” and “it’s very dangerous to put a person in a splint they’re not prepared for.” Why? Does the available literature deal with this?

Conclusion

Concise and relevant to the study findings.

Limitations:

Well-articulated and identifies the four points of limitation clearly.

Recommendations:

These are very useful suggestions for addressing the problems identified in the study. I hope that at least some of these well thought out and useful ideas will be implemented.

Referencing:

Sequence of numbering of citations starts being chronologically inconsistent on Page 12 after reference 17. I’m not sure citation 18 is included in the manuscript. Please check.

**Do you want your identity to be public for this peer review?** For information about this choice, including consent withdrawal, please see our Privacy Policy

Reviewer #1: No

Reviewer #2: No

Reviewer #3: No

Reviewer #4: No

---

## [Author Response · Author response to Decision Letter 1]

25 Sep 2025

Thank for all the detailed feedback.

---

## [Decision Letter · Decision Letter 1]

19 Nov 2025

Dear Dr.  Monareng,

We look forward to receiving your revised manuscript.

Kind regards,

Ahmed Abdelwahab Ibrahim El-Sayed

Academic Editor

PLOS ONE

Journal Requirements:

Additional Editor Comments:

Dear Authors,

Thank you for your contribution.

The reviewers have identified several minor issues that still need to be carefully addressed before we can proceed to a final editorial decision regarding your manuscript. Please review their comments thoroughly and make the necessary revisions

Reviewer's Responses to Questions

**Comments to the Author**

Reviewer #3: (No Response)

Reviewer #5: (No Response)

Reviewer #6: All comments have been addressed

2. Is the manuscript technically sound, and do the data support the conclusions?

Reviewer #3: Yes

Reviewer #5: Partly

Reviewer #6: Yes

3. Has the statistical analysis been performed appropriately and rigorously?

Reviewer #3: N/A

Reviewer #5: I Don't Know

Reviewer #6: No

4. Have the authors made all data underlying the findings in their manuscript fully available?

Reviewer #3: Yes

Reviewer #5: Yes

Reviewer #6: Yes

5. Is the manuscript presented in an intelligible fashion and written in standard English?

Reviewer #3: Yes

Reviewer #5: Yes

Reviewer #6: Yes

Reviewer #3: Your revised manuscript presents a valuable and well-executed qualitative study on occupational therapy student preparedness for splinting in hand rehabilitation. You have addressed previous concerns thoughtfully, and the revised version significantly improves clarity and academic rigor. Only minor issues remain regarding the clarity of the language and some methodological elaboration.

The revised version shows clear improvements in grammar, tone, and structure.

The manuscript now reads more fluently, and redundancies have been reduced. However, occasional inconsistency in tense (e.g., shifting between present and past when describing methods or context). A language polish would ensure clarity and readiness for publication.

The authors may wish to mention implications for global OT education, especially in low-resource or similar healthcare settings, to enhance relevance.

Line 324: "Data saturation was achieved..." – consider briefly stating how.

Reviewer #5: This ia a thoughtful exploration of a perceived problem of hand splinting curriculum in OT training.

Lines 172-4 "traumatic hand injuries, accounting for 17.7% of all cases in KZN" and lines 668-90 "traumatic hand injuries, which constitute approximately 17.7% of cases in KZN" both misquote the source ref Naidoo, who actually states "In the province of KwaZulu-Natal (KZN) in South Africa (SA), trauma is extensive, constituting at least 17.8% of overall emergency cases", and who references the 17.8% statistic to a paper by Stewart that itself does not mention hand trauma. This wording is misleading.

Line 336. Please define reflexivity.

The description of the participant selection process lacks sufficient detail. What was the population size from which the participants were selected? What were the specifics of the purposive sampling strategy? What measures did the authors take to avoid sampling bias from recruiting a nonrepresentative proportion of disgruntled students - for example, what wording was used to describe the purpose of the study to potential participants?

Because of the repeated statements that inadequate time was devoted to hand splint instruction, please provide context to compare what proportion of a typical starting OT practice would be expected to involve hand splinting with what proportion of the current OT curriculum is devoted to hand splinting.

It might be assumed from the manuscript that the only instruction received on hand splinting was in person. What instructional reference materials were provided to students regarding indications and techniques of hand splinting?

Reviewer #6: The authors made extensive revisions to address all comments from reviewers. This study explains an important aspect of lack of resources in South Africa and also limited training for students seeking to address this shortage of resources. This is a common issue in the entire world, but is explained in this paper for South Africa.

I think a re-work of the abstract to more clearly and simply explain the lack of preparedness that OT students feel through training is important. There are some aspects of the training that I think the authors assume the readers will know - for instance, how many years is this training supposed to be? It appears to be 4, but it is not clear to me. Different countries may have varying school lengths. Also, I am not clear on what "blocks" are, which are described in the abstract and in the paper itself.

For more specifics, the formatting of the study responses (italics) in the paper helps delineate it from the rest of the paper, but making these sections isolated in table-form, may make reading the paper more easy.

**Do you want your identity to be public for this peer review?** For information about this choice, including consent withdrawal, please see our Privacy Policy

Reviewer #3: No

Reviewer #5: No

Reviewer #6: No

---

## [Author Response · Author response to Decision Letter 2]

20 Nov 2025

Please refer to the attached document titled "Rebuttal letter Responses to academic editor and reviewers"

---

## [Decision Letter · Decision Letter 2]

4 Jan 2026

Thank you for submitting your manuscript to PLOS ONE. After careful consideration, we feel that it has merit but does not fully meet PLOS ONE’s publication criteria as it currently stands. Therefore, we invite you to submit a revised version of the manuscript that addresses the points raised during the review process.

We look forward to receiving your revised manuscript.

Kind regards,

Ahmed Abdelwahab Ibrahim El-Sayed

Academic Editor

PLOS One

Journal Requirements:

Additional Editor Comments:

Dear Authors,

Thank you for submitting your manuscript to PLOS ONE. We appreciate the time and effort you have invested in this work.

Based on the reviewers’ evaluations, there are a number of minor comments that require your attention and should be resolved prior to reaching a final decision.

Reviewers' comments:

Reviewer's Responses to Questions

**Comments to the Author**

Reviewer #5: All comments have been addressed

Reviewer #7: (No Response)

Reviewer #8: All comments have been addressed

2. Is the manuscript technically sound, and do the data support the conclusions?

Reviewer #5: Yes

Reviewer #7: Yes

Reviewer #8: Yes

3. Has the statistical analysis been performed appropriately and rigorously?

Reviewer #5: I Don't Know

Reviewer #7: N/A

Reviewer #8: N/A

4. Have the authors made all data underlying the findings in their manuscript fully available?

Reviewer #5: Yes

Reviewer #7: Yes

Reviewer #8: Yes

5. Is the manuscript presented in an intelligible fashion and written in standard English?

Reviewer #5: Yes

Reviewer #7: Yes

Reviewer #8: Yes

Reviewer #5: I have one remaining request. In my prior review, I stated:

Lines 172-4 "traumatic hand injuries, accounting for 17.7% of all cases in KZN" and lines 668-90 "traumatic hand injuries, which constitute approximately 17.7% of cases in KZN" both misquote the source ref Naidoo, who actually states "In the province of KwaZuluNatal (KZN) in South Africa (SA), trauma is extensive, constituting at least 17.8% of overall emergency cases", and who references the 17.8% statistic to a paper by Stewart that itself does not mention hand trauma. This wording is misleading.

The authors responded:

Lines 172-4: Global trends indicate an increasing number of traumatic hand injuries [19], while just above 30% of traumatic injuries are hand-related in South Africa [20].

However, reference 19 (Stewart, 2017) does not mention trends in the number of hand injuries. Please either modify the wording or supply a reference which supports the statement of increasing numbers.

Reviewer #7: Major Comments

Experiential learning sequencing and confidence development in splinting education

This manuscript presents a technically sound qualitative exploration of undergraduate occupational therapy students’ perceptions of preparedness for splinting in hand rehabilitation. The study is methodologically appropriate, ethically conducted, and clearly reported. The results are well supported by participant quotations and align with existing literature regarding the perceived gap between theoretical instruction and practical confidence. The following comments are offered to strengthen interpretation and discussion rather than to suggest changes to study design or scope.

Several findings indicate that students’ limited confidence in splinting may be influenced not only by the quantity of practical exposure, but also by how experiential learning opportunities are sequenced and reinforced over time. While participants frequently describe insufficient practice time or limited clinical exposure, the data suggest that learning opportunities may be episodic and unevenly structured, which may inhibit consolidation of procedural competence.

The discussion could be strengthened by more explicitly situating the findings within a staged experiential learning process consistent with Kolb’s experiential learning theory, which the authors reference as a guiding framework. Participants appear to engage in reflective observation and abstract conceptualization through theoretical instruction, yet report limited opportunities for structured active experimentation across progressively complex contexts. This partial traversal of the experiential learning cycle may help explain why increased exposure alone does not consistently translate into confidence or perceived readiness.

Early stages of skill development may benefit from deliberate practice under low cognitive load, emphasizing repetition, immediate feedback, and error tolerance to support initial procedural encoding. As skills begin to consolidate, learners may require contextualized practice that introduces controlled human variability and tactile feedback, enabling translation from procedural familiarity to applied competence. Finally, learning experiences that approximate clinical complexity may support integration of technical execution with communication, situational awareness, and adaptive decision-making, thereby completing the experiential learning cycle.

Relatedly, several participants describe fluctuations or erosion of confidence between instructional or clinical blocks, suggesting that confidence development may be sensitive to continuity and reinforcement of practice. Episodic exposure without sufficient opportunities for consolidation may result in partial skill acquisition, leaving learners theoretically prepared but uncertain in execution. Framing confidence as a dynamic construct influenced by reinforcement and skill consolidation over time may provide a more nuanced interpretation of participant recommendations for increased practice time.

Incorporating these perspectives into the discussion would allow the authors to interpret calls for additional practice and curriculum expansion as a need for structured progression and longitudinal reinforcement, rather than solely increased exposure. This framing remains consistent with the qualitative findings and theoretical positioning of the study while strengthening its educational implications and transferability to other practice-oriented health professions programs.

Minor Comments

1. Clarification of analytic approach

The methods section notes the use of thematic analysis guided by COREQ and references both inductive and deductive elements. A brief clarification of how deductive components were operationalized, and how inductive coding was protected from excessive theoretical constraint, may enhance transparency for readers less familiar with hybrid qualitative approaches.

2. Sample sufficiency and interpretive scope

The results are supported by rich participant quotations. Where appropriate, the authors may consider briefly clarifying whether the intent of the sample size was to achieve thematic saturation or depth of perspective, particularly given the exploratory nature of the study.

3. Alignment of recommendations and themes

The discussion may be further strengthened by explicitly linking participant-recommended solutions to specific themes, clarifying whether perceived gaps are interpreted primarily as curricular sequencing issues, clinical placement variability, or an interaction of both.

Reviewer #8: Dear Authors, thank you for your efforts in improving the presentation of the paper.

It is a valuable paper, as it delves into the issues surrounding competency-based training.

**Do you want your identity to be public for this peer review?** For information about this choice, including consent withdrawal, please see our Privacy Policy

Reviewer #5: No

Reviewer #7: No

Reviewer #8: No

---

## [Author Response · Author response to Decision Letter 3]

22 Jan 2026

The authors did not receive funding for this research.

---

## [Editor Report · Decision Letter 3]

1 Feb 2026

Undergraduate occupational therapy students’ perceptions of their preparedness for splinting in hand rehabilitation: an exploratory study at the University of KwaZulu-Natal

PONE-D-25-01929R3

Dear Author,

We’re pleased to inform you that your manuscript has been judged scientifically suitable for publication and will be formally accepted for publication once it meets all outstanding technical requirements.

Kind regards,

Ahmed Abdelwahab Ibrahim El-Sayed

Academic Editor

PLOS One

Additional Editor Comments :

Dear authors,

Thank you for your efforts to enhance your manuscript. I can accept your study in its current form for publication at PLOS ONE. CONGRATULATIONS

---

## [Editor Report · Acceptance letter]

PONE-D-25-01929R3

PLOS One

Dear Dr. Monareng,

I'm pleased to inform you that your manuscript has been deemed suitable for publication in PLOS One. Congratulations! Your manuscript is now being handed over to our production team.

Kind regards,

on behalf of

Dr. Ahmed Abdelwahab Ibrahim El-Sayed

Academic Editor

PLOS One